# *p*-Coumaric acid, Kaempferol, Astragalin and Tiliroside Influence the Expression of Glycoforms in AGS Gastric Cancer Cells

**DOI:** 10.3390/ijms23158602

**Published:** 2022-08-02

**Authors:** Iwona Radziejewska, Katarzyna Supruniuk, Michał Tomczyk, Wiktoria Izdebska, Małgorzata Borzym-Kluczyk, Anna Bielawska, Krzysztof Bielawski, Anna Galicka

**Affiliations:** 1Department of Medical Chemistry, Faculty of Pharmacy, Medical University of Białystok, ul. Mickiewicza 2a, 15-222 Białystok, Poland; katarzyna.supruniuk@umb.edu.pl (K.S.); izdebskawiktoria@gmail.com (W.I.); anna.galicka@umb.edu.pl (A.G.); 2Department of Pharmacognosy, Faculty of Pharmacy, Medical University of Białystok, ul. Mickiewicza 2a, 15-222 Białystok, Poland; michal.tomczyk@umb.edu.pl; 3Department of Pharmaceutical Biochemistry, Faculty of Pharmacy, Medical University of Białystok, ul. Mickiewicza 2a, 15-222 Białystok, Poland; gocha@umb.edu.pl; 4Department of Biotechnology, Faculty of Pharmacy, Medical University of Białystok, ul. Kilińskiego 1, 15-089 Białystok, Poland; aniabiel@umb.edu.pl; 5Department of Synthesis and Technology of Drugs, Faculty of Pharmacy, Medical University of Białystok, ul. Kilińskiego 1, 15-089 Białystok, Poland; kbiel@umb.edu.pl

**Keywords:** gastric cancer, glycosylation, MUC1, polyphenolics

## Abstract

Abnormal glycosylation of cancer cells is considered a key factor of carcinogenesis related to growth, proliferation, migration and invasion of tumor cells. Many plant-based polyphenolic compounds reveal potential anti-cancer properties effecting cellular signaling systems. Herein, we assessed the effects of phenolic acid, *p*-coumaric acid and flavonoids such as kaempferol, astragalin or tiliroside on expression of selected cancer-related glycoforms and enzymes involved in their formation in AGS gastric cancer cells. The cells were treated with 80 and 160 µM of the compounds. RT-PCR, Western blotting and ELISA tests were performed to determine the influence of polyphenolics on analyzed factors. All the examined compounds inhibited the expression of MUC1, ST6GalNAcT2 and FUT4 mRNAs. C1GalT1, St3Gal-IV and FUT4 proteins as well as MUC1 domain, Tn and sialyl T antigen detected in cell lysates were also lowered. Both concentrations of kaempferol, astragalin and tiliroside also suppressed ppGalNAcT2 and C1GalT1 mRNAs. MUC1 cytoplasmic domain, sialyl Tn, T antigens in cell lysates and sialyl T in culture medium were inhibited only by kaempferol and tiliroside. Nuclear factor NF-κB mRNA expression decreased after treatment with both concentrations of kaempferol, astragalin and tiliroside. NF-κB protein expression was inhibited by kaempferol and tiliroside. The results indicate the rationality of application of examined polyphenolics as potential preventive agents against gastric cancer development.

## 1. Introduction

Gastric cancer (GC) is a major health burden worldwide and one of the leading causes of cancer deaths. Very poor outcomes of GC are related with its diagnosis at advanced stages because of an asymptomatic character at the early stages. Complete surgical resection seems to be the only opportunity for cure in GC [1]. However, in recent decades, a gradual decrease in GC rates has been observed in many countries. It is stated that such a decrease reflects some trends in food handling, among others, the abundance of fresh fruit and vegetables in the diet [2]. Many plant-based compounds, including polyphenols, reveal anti-cancer properties, without being toxic for healthy cells [3]. It has been reported that polyphenols can affect several cancer preventive mechanisms such as prevention of oxidation, induction of apoptosis, anti-inflammatory properties and effects on the cellular signaling systems [4]. Thus, such compounds represent a great promise in cancer treatment, including GC. 

Specific glycosylation alterations of proteins, especially *O*-glycosylation of mucins, main glycoproteins of stomach mucosa, are, among others, less-studied molecular events accompanying cancer development. Glycans can disturb cell–cell and cell–extracellular matrix adhesion features reinforcing cancer cell migration, invasion and metastasis [5,6,7]. Expression of truncated simple *O*-glycans (TACAs—tumor-associated carbohydrate antigens) and increase in sialylation and fucosylation are said to be major events involved in oncogenic transformations in GC [8]. It has been also stated that such changes in protein glycosylation definitely contribute to the development of more malignant characteristics of cancer [9].

Our recent promising results concerning the anti-cancer potential of rosmarinic acid (polyphenolic compound isolated from medicinal species of Boraginaceae and Lamiaceae such as rosemary or mint) by glycosylation modifications in AGS gastric cancer cells [10,11] raise the question of whether other polyphenolic compounds could reveal such potential. Thus, in our study we decided to assess the effects of selected polyphenolic compounds—phenolic acid, *p*-coumaric acid and derivatives of flavonols such as kaempferol, astragalin and tiliroside (Figure 1)—on specific glycoforms’ expression in AGS gastric cancer cells. We included Tn, T, sialyl Tn, sialyl T and fucosylated carbohydrate structures, glycosyltransferases participating in formation of mentioned antigens as well as MUC1 mucin, main carrier of such antigens. 

## 2. Results

### 2.1. Viability of AGS Gastric Cancer Cells

To assess the effects of coumaric acid, kaempferol, astragalin and tiliroside on the viability of gastric cancer cells, MTT test was applied. Used 20–160 μM concentrations of the examined compounds revealed low cytotoxic effects. Viability of the cells was higher than 80% at all concentrations compared with untreated control cells (with the exception of kaempferol with 160 µM concentration, where viability was equal 80%) (Figure 2). In all the experiments in the study, we decided to use 80 and 160 μM concentrations of the polyphenolics. 

### 2.2. The Effect of the Compounds on MUC1 Expression

In the study, RT-PCR was used to determine MUC1 mRNA, and Western blotting was applied for assessment of MUC cytoplasmic domain, MUC1 in cell lysates and MUC1 extracellular domain released to the culture medium. The cells were incubated with 80 and 160 μM coumaric acid, kaempferol, astragalin and tiliroside for 24 h. MUC1 mRNA expression was significantly inhibited after action of all the compounds, with the strongest effect after astragalin and tiliroside action (Figure 3A). As MUC1 is a glycoprotein with a subunit structure, it can be detected in the form of a cytoplasmic tail, a domain which is membrane-tethered (detected in cell lysates) and an extracellular domain released to the culture medium. Both concentrations of kaempferol and tiliroside as well as 160 µM coumaric acid significantly diminished the expression of MUC1 cytoplasmic tail (Figure 3B). All the applied polyphenolics suppressed MUC1 in cell lysates (Figure 3C). The changes were statistically significant. On the contrary, rather low effect of the chemicals on the extracellular domain of the mucin released to the culture medium was observed (Figure 3D). This part of MUC1 was revealed in the form of two bands, with approximately 245 kDa and 180 kDa molecular masses. The expression of the domain with higher molecular mass was significantly inhibited only by 160 μM concentration of coumaric acid and kaempferol, and 180 kDa domain by the same concentration of tiliroside. 

### 2.3. The Effect of the Compounds on Tn and sialyl Tn Antigen Expression

Tn antigen (GalNAcα1-O-Ser/Thr) is formed after attaching GalNAc to serine or threonine of protein polypeptide chain in the reaction catalyzed by ppGalNAcT2 (polypeptide N-acetyl-α-galactosaminyltransferase). This antigen is the simplest sugar antigen from which other carbohydrate forms are created after the action of proper enzymes. RT-PCR analysis revealed significant suppression of ppGalNAcT2 mRNA expression as a result of kaempferol, astragalin and tiliroside action in comparison with untreated control (Figure 4A). All sugar antigens examined in the study can be present on the cell surfaces as well as on the glycoproteins released to extracellular matrix. Due to this, we decided to examine the presence of such antigens in cell lysates and culture medium. ELISA test with VVA biotinylated lectin (specification of all used lectin is presented in Table 1) with high affinity to Tn sugar structure showed statistically significant inhibition of this antigen in cell lysates by 80 μM coumaric acid, and both used concentrations of kaempferol, astragalin and tiliroside (Figure 4B). The strongest effect was observed for kaempferol and tiliroside.

Sialylated form of Tn antigen is created upon action of proper sialyltransferases such as ST6GalNAcT2. In Figure 4C, we can observe significant inhibition of ST6GalNAcT2 mRNA expression by all tested compounds, with the lowest effect revealed by coumaric acid. Biotinylated SNA lectin was applied in an ELISA test to detect sialyl Tn antigen in cell lysates and culture medium. 80 and 160 µM concentrations of kaempferol and tiliroside significantly decreased the level of this structure in cell lysates. In culture medium, such effect, however less intensified, was observed after 160 μM kaempferol and both concentrations of astragalin and tiliroside treatment (Figure 4D). 

### 2.4. The Effect of the Polyphenolic Compounds on T and sialyl T Antigen Expression

Core 1 β1,3-galactosyltransferase (C1GalT1) is an enzyme catalyzing T antigen formation (Galβ1,3-GalNAc-O-Ser/Thr) by transferring galactose on Tn antigen. Both concentrations of kaempferol, astragalin and tiliroside significantly suppressed the expression of C1GalT1 mRNA (Figure 5A). The strongest effect can be observed for kaempferol and tiliroside. 160 µM of coumaric acid slightly increased C1GalT1 expression. Western blotting revealed that C1GalT1 protein was significantly inhibited by all the examined polyphenolics with the exceptions of 160 µM astragalin (Figure 5B). ELISA test performed with PNA lectin with the affinity to T antigen discovered that the expression of T antigen in cell lysates had similar tendency of changes as C1GalT1 mRNA expression (Figure 5C). T antigen in the culture medium was only slightly diminished by the polyphenolic compounds. 

The next examined carbohydrate structure was sialyl T antigen (SAα2,3-Galβ1,3-GalNAc-O-Ser/Thr) detected with MAAII biotinylated lectin. The reaction of this antigen formation is catalyzed by Gal α2,3-sialyltransferase (ST3GalT1). Both concentrations of kaempferol and astragalin, as well as 160 μM coumaric acid and tiliroside, significantly inhibited ST3GalT1 mRNA expression (Figure 5D). Surprisingly, 80 μM coumaric acid stimulated this gene expression. St3Gal-IV protein was significantly suppressed by all applied compounds (Figure 5E). In the ELISA test, we revealed that all polyphenolics inhibited the expression of T antigen in cell lysates, with the strongest effect after kaempferol and tiliroside action (Figure 5F). All the changes were statistically significant. In culture medium, a rather modest, decreasing effect was observed after kaempferol and tiliroside action as well as 160 μM astragalin. 

### 2.5. The Effect of Compounds on Fucosylated Antigen Expression

α1,3-fucosyltransferase (FUT4) is the enzyme responsible for Fucα1,3-GalNAc structure formation. FUT4 mRNA as well as FUT4 protein were significantly diminished by the action of all compounds used in the study (Figure 6A,B). LTA biotinylated lectin, used in the ELISA test, allowed to Fucα1,3-GalNAc antigen in cell lysates and culture medium of gastric cancer cells to be detected. Coumaric acid, kaempferol, tiliroside with 80 and 160 μM and astragalin with 160 μM significantly diminished the expression of Fucα1,3-GalNAc antigen in cell lysates. Kaempferol and tiliroside revealed stronger inhibitory effect than coumaric acid and astragalin (Figure 6C). In the case of the ELISA test performed with culture medium, slightly lower expression of tested antigen was observed only for 160 μM kaempferol.

### 2.6. The Effect of the Compounds on NF-κB Expression

The last factor assessed in the study, inherently connected with carcinogenesis, was nuclear transcription factor NF-κB. We assumed that NF-κB signaling pathway can be involved in inhibition of factors engaged in glycosylation process. Kaempferol, astragalin, tiliroside and 160 μM coumaric acid strongly inhibited NF-κB expression (Figure 7A). The changes were statistically significant. However, NF-κB protein was suppressed only by kaempferol and tiliroside (Figure 7B).

## 3. Discussion

It has been reported that many polyphenolic compounds exhibit anti-tumor functions by controlling cell viability or inducing apoptosis. Their application in cancer treatment gains attention due to their safety, efficiency and low side effects [12,13,14,15,16,17,18]. Altered glycosylation is also considered as a hallmark of carcinogenesis with involvement in the disease outcome [19]. Abnormal glycosylation is claimed to be a key factor of tumor malignant transformation and is closely related to the biological behaviors of tumor cells, such as growth, proliferation, migration and invasion [13,20]. However, because of the complexity and heterogeneity of glycan structures, full understanding of the role protein glycosylation in carcinogenesis remains a big challenge in the field. Thus, due to the limited studies concerning the effects of polyphenolics on glycosylation processes, we decided to evaluate the influence of selected polyphenolic compounds on such events in AGS gastric cancer cells. Coumaric acid, kaempferol, astragalin and tiliroside were involved in the study.

*p*-Coumaric acid (4-hydroxycinnamic acid) (Figure 1A) is phenolic acid found ubiquitously in plants and mushrooms, in free or bound forms, revealing various bioactivities, including anti-cancer property. It can be subsequently transformed to other flavonoids [14]. Kaempferol (Figure 1B), a flavonoid widely present in many fruits, vegetables and traditional herbal medicines, has been reported as a cancer therapeutic agent by its anti-proliferative action, promoting autophagy and apoptosis [15,16]. Astragalin (kaempferol-3-*O*-β-D-glucoside) (Figure 1C), another naturally occurring flavonoid, polyphenolic compound, was demonstrated to induce cell death and inhibit proliferation and migration of cancer cells [17]. The last flavonoid examined in this study was tiliroside (kaempferol-3-β-D-(6″-*O*-*p*-coumaroyl)-glucopyranoside) (Figure 1D), with anti-cancer potential revealed by affecting apoptosis and metastasis of cancer cells [18,21].

As mentioned above, pointing out aberrant glycosylation as a therapeutic destination in cancer therapy seems to be a promising strategy. The main glycoprotein of gastric epithelium is MUC1 mucin. Its overexpression and specifically altered glycosylation have been associated with cancer progression and metastasis that correlated with poor prognosis and high death rate [19,22,23,24,25]. The National Cancer Institute Translational Research Working Group listed this mucin as one of the most promising targets in cancer research [26]. MUC1 is composed of two subunits, a long, highly *O*-glycosylated N-terminal MUC1-N fragment that is non-covalently bound with a short C-terminal MUC1-C part. Within the MUC1-C subunit, there is a short extracellular domain, a transmembrane domain and a cytoplasmic tail [22,27]. Both MUC1-N and MUC1-C subunits are considered to be important in cancer development [27]. Heavy glycosylated MUC1-N part can participate in various glycan–protein recognition events that may contribute to tumor progression [28,29]. The membrane-tethered part can act as a membrane receptor taking part in regulation of several inflammatory processes through phosphorylation of MUC1-C cytoplasmic tail [30]. Thus, MUC1 may activate and mediate intracellular signaling by associating with various protein receptors and modulating their functions leading to altered signaling in cancer cells [22]. There are also reports about the primary role of defective *O*-glycosylation in the pathogenesis of gastritis-associated cancers [7].

Some authors demonstrated that selected flavonoids were able to inhibit tumorigenesis by inhibition of MUC1 expression. Wang et al. revealed such action of quercetin in breast cancer cells [31]. Moreover, Zhou et al [32] noticed suppression of MUC1 mRNA and MUC1 cytoplasmic tail by apigenin in MCF-10A cells. Recently, we showed also that luteolin decreased expression of MUC1 extracellular domain in CRL-1739 gastric cancer cells [33]. The results of this study support the mentioned outcomes as we noticed inhibition of MUC1 mRNA and MUC1 glycoprotein after the action of all examined polyphenolics. Cytoplasmic tail expression was also suppressed but only by kaempferol, tiliroside as well as by 160 µM of coumaric acid. According to this, we postulate the polyphenolic compounds applied in the study can act as potential anti-cancer agents because of inhibitory effects on MUC1 oncoprotein.

The next aim of our experiments was to assess how coumaric acid, kaempferol, astragalin and tiliroside influence selected tumor-associated antigens (TACAs) and the enzymes taking part in their formation. Such antigens, formed in cancer cells because of aberrant glycosylation, have been established by the National Institute of Health as significant biomarkers of cancer prognosis [34]. They are typical for tumors, but not for normal cells [6]. They are associated with the promotion of tumor cell invasion, metastasis and, due to their immunogenicity, are considered to be unique targets for cancer vaccine design and development [27,35]. The most common TACAs formed from incomplete synthesis are Tn (GalNAcα1-O-Ser/Thr), T (Galβ1,3-GalNAc-O-Ser/Thr), sialyl Tn (SA. α2,6-GalNAcα1-O-Ser/Thr) and sialyl T (SAα2,3-Galβ1,3-GalNAc-O-Ser/Thr). Formation of listed structures is catalyzed by specific glycosyltransferases such as polypeptide N-acetyl-α-galactosaminyltransferase (e.g., ppGalNAcT2), core 1 β1,3-galactosyltransferase (C1GalT1), α2,6-sialyltransferase (ST6GalNAcT) and α2,3-sialyltransferase (ST3GalT) respectively [6]. Overexpression of fucosylated structures is also typical for many cancers. They are considered as tumor biomarkers involved in increased proliferation, invasion and metastatic capacity. One such structure is Fucα1,3-GalNAc, whose formation is catalyzed by α1,3-fucosyltransferase (FUT4) [36]. There have been some reports that flavonoids influence the glycosylation process in cancers. Liu et al [37] demonstrated that tannin-based polyphenols suppressed ppGalNAc-Ts activity in colorectal cancer and suggested such dietary product as the factor disturbing *O*-glycosylation and acting as a cancer-preventing agent by inhibition of migration and invasion. Huang et al [13] has proposed flavonoid eriodictyol as a promising anti-cancer agent due to restraining fucosylation in CRC cells. Recently, we suggested afzelin (kaempferol 3-*O*-α-L-rhamnopyranoside) as the structure suppressing glycosylation (e.g., by inhibition of C1GalT1, ST6GalNAcT, ST3GalT expression) in gastric cancer cells and in this way acting as a potential anti-cancer agent [38]. We demonstrated also that rosmarinic acid decreased expression of Tn, T antigen and their sialylated forms in gastric cancer cells [10]. In our opinion, the current results presented in this study support the ideas discussed above considering natural flavonoids as anti-cancer agents. We observed the changes in TACAs expression in cell lysates and in culture medium. Such glycoforms can be present at cell surfaces as well as on glycoproteins (such as MUC1) released to extracellular matrix; they can be aberrantly overexpressed during cancer development in both localizations [8,22,23]. We conclude that the inhibition of specific glycoforms expression revealed in our study could be treated as the preventive treatment against cancer development.

At the last point of our experiments, we considered the influence of coumaric acid, kaempferol, astragalin and tiliroside on NF-κB expression. NF-κB signaling pathway activation is stated as a pivotal player involved in cell proliferation, cell death, cell invasion and metastasis in many cancers, and due to this it is considered as a molecular target in malignancies [39]. There are many reports about the suppressing of this factor by naturally occurring compounds. Chen et al [40] revealed MAPKs and NF-κB activity inhibition by astragalin in NSCLC cells. Similar action of astragalin was documented by Yang et al. in HCT116 colon cancer cells [17]. Several other phytochemicals have been proposed to possess capabilities to suppress cancer cell proliferation by blocking NF-κB nuclear translocation or reducing its activation [41,42]. According to our results, we assume that kaempferol and tiliroside can be added to the group of mentioned compounds with abilities to inhibit NF-κB on mRNA and protein level. We revealed that coumaric acid and astragalin suppressed only NF-κB mRNA, and there was no effect on nuclear factor protein. We can suggest that NF-κB signaling pathway can be involved in inhibition of factors as glycosyltransferases involved in modulation of glycosylation process. Sokolova et al. [43] proposed NF-κB as an agent that regulates many genes promoting gastric carcinogenesis. Fu et al. [44] provided evidence that sialyltransferases’ inhibition by NF-κB signaling pathway reduces breast cancer cell metastasis. Other authors also noted relationships between NF-κB-mediated signaling with glycosylation changes in cancers [8].

To summarize, we suggest that the four polyphenolic compounds examined in the study—coumaric acid, kaempferol, astragalin and tiliroside—reveal anti-cancer potential and can be considered as a kind of prevention agents against gastric cancer development. Such a conclusion is stated on the results demonstrating inhibition of specific glycoforms expressions, structures which are proved to participate in cancer development. Summarized inhibitory effects are presented in Figure 8.

According to our results, kaempferol, astragalin and tiliroside seem to reveal higher activity towards examined factors than *p*-coumaric acid, which is in accordance with Pei et al. [14], who suggested that activity of coumaric acid increases after the formation of conjugates. It has been also demonstrated that tiliroside bearing *p*-coumaroyl moiety exhibited higher antioxidant and cytoprotective effect than astragalin [45]. On the other hand, astragalin and tiliroside have a kaempferol moiety and we can suggest that this basic structure can be responsible of the activity of the compounds. From the outcomes of the present study, we can also conclude that higher anti-cancer potential of kaempferol, astragalin and tiliroside can be connected with the appearance of −OH groups in their structures. However, such conclusion should be proved by other experiments, which are planned to be performed in the future. We are also aware of limitations concerning validation of MUC1 as a main carrier of the identified O-glycans. A mass spectrometry-based O-glycomics approach of MUC1 electrophoretic bands could prove such a claim. Moreover, a future understanding of the correlation of MUC1 carrying specific carbohydrate antigens with cellular adhesion, invasion capacity and potential resistance to therapeutic options is also required.

## 4. Materials and Methods

### 4.1. Polyphenolic compounds

*p*-Coumaric acid was purchased from Sigma-Aldrich (Steinheim, Germany). Kaempferol was purchased from Carl Roth GmbH (Karlsruhe, Germany). Astragalin (kaempferol 3-*O*-β-D-glucopyranoside) and tiliroside (kaempferol 3-*O*-β-D-(6″-*E*-*p*-coumaroyl)-glucopyranoside) were isolated from flowers of *Ficaria verna* Huds. (Ranunculaceae) and leaves of *Rubus caesius* L. (Rosaceae), respectively. Their chemical structures were identified by using spectral (UV, NMR, MS) methods. All isolated compounds were >96% pure, as measured by HPLC [46,47].

### 4.2. Cell Culture

CRL-1739 human gastric adenocarcinoma cells (AGS) were obtained from American Type Culture Collection (ATCC, Manassas, VA, USA). They were cultured in F-12 medium (Gibco, Waltham, MA, USA) in a humidified atmosphere of 5% CO_2_ at 37 °C. The medium was enriched with streptomycin (100 μg/mL), penicillin (100 U/mL) (Sigma, St. Luis, MO, USA) and 10% Fetal Bovine Serum (FBS) (Gibco, Waltham, MA, USA). Then, the cells were seeded into 6-well plates (about 5 × 10^5^ cells/well) and cultured for 24 hours in 1 mL of growth medium (FBS-free) supplemented with 80 and 160 μM of coumaric acid, kaempferol, astragalin or tiliroside. Stock solutions of the added compounds were 160 mM (prepared in DMSO; Sigma, St. Luis, MO, USA). Next, the cells were washed with PBS (Phosphate Buffered Saline, pH 7.4) and lysed for 20 min at 4 °C with an RIPA buffer (Sigma, St. Luis, MO, USA) enriched with protease inhibitors cocktail (Sigma, St. Luis, MO, USA) diluted 1:200 (in RIPA buffer). The collected lysates and culture media were centrifuged at 1000× *g* for 5 min at 4 °C. The supernatants were frozen at −70 °C and used for Western blot and ELISA analyses. For RT-PCR assays, the monolayers were washed three times with sterile PBS (10 μM) and sonificed (Sonics Vibra cell; Sonics & Materials, Leicestershire, UK) to disrupt the cell membranes. Aliquots of homogenate were used to isolate RNA. Cells without added compounds were taken as control.

### 4.3. Cell Viability Test

The measurements of the viability of the cultured cells, in the presence of coumaric acid, kaempferol, astragalin and tiliroside, were assayed using 3-(4,5-dimethylthiazole-2-yl)-2,5-diphenyltetrazolium bromide (MTT) (Sigma, St. Louis, MO, USA) according to the procedure of Carmichael et al [48]. Briefly, cells were cultured in 6-well plates to achieve 70% of confluency and then they were incubated for 24 h with 0–160 μM concentrations of the compounds mentioned above. After that, there was incubation of the cells in 1 mL of MTT solution (0.5 mg MTT/mL PBS) at 37 °C in 5% CO_2_ for 4 h. The measurements of absorbance of converted dye at 570 nm was used. The viability of gastric cancer cells was determined as a percentage of control cells (100% cell viability).

### 4.4. RNA Isolation and RT-PCR

Total RNA was isolated applying Total RNA Mini Plus Concentrator (A&A Biotechnology, Gdansk, Poland), according to the instruction provided. RNA purity and concentration was assessed spectrophotometrically (Nanodrop 2000, Thermo Scientific, Waltham, MA, USA). Equal amounts (1 μg) of total RNA were subjected to reverse transcription using the SensiFASTTM cDNA Synthesis Kit (Bioline, London, UK). Twenty microliters of the reaction mixture contained RNA template, one microliter of Reverse Transcriptase and four microliters of 5xTransAmp Buffer and DEPC-treated water. Ten minutes at 25 °C, thirty minutes at 45 °C, and five minutes at 70 °C were conditions of incubation. cDNA amplification was performed using SensiFAST™ SYBR Kit (Bioline, London, UK) in the thermocycler CFX96 real-time system (BioRad, Hercules, CA, USA). Reaction mixture (total volume 20 μL) contained 2 μL of cDNA template (3-times diluted), 0.4 μL of each target-specific primer (10 μmol/L) (Genomed, Warsaw, Poland) (Table 2), 2×SensiFAST SYBR No-ROX Mix (5 μL) and DEPC-treated water. Glyceraldehyde-3-phosphate dehydrogenase (GAPDH) was used as a reference gene. The PCR plate was subjected to the following reaction parameters: 95 °C for 2 min (activation of DNA polymerase), followed by 40 cycles of 10 s at 95 °C (denaturation), 15 s at 60 °C (annealing) and elongation (20 s at 72 °C). Each sample was examined in triplicate. To confirm the formation of a single product, the melting curves were analyzed after amplification. The levels of target gene transcripts were normalized to GAPDH transcripts using the ∆∆Ct method.

### 4.5. Western Blotting

Electrophoresis on 7.5–13% polyacrylamide gel and Western blotting analysis were performed to detect the expression of MUC1, C1GalT1, St3Gal-IV, FUT4, NF-κB in cell lysates as well as MUC1 extracellular domain in culture media. The samples (20 μg of protein) were diluted 4:1 in a probe sample buffer containing 2.5% SDS (Sigma, St. Luis, MO, USA), subjected to electrophoretic separation and transferred to an Immobilon P membrane (Millipore, Bedford, MA, USA) according to the method of Towbin et. al [49]. Five percent skim milk in Tris Buffered Saline, pH 7.4 (TBS) with 0.05% Tween 20 (Sigma, St. Luis, MO, USA), was applied to block the membranes for 1 h at room temperature (RT). After washing step in TBS-T, the membranes were incubated with respected primary monoclonal antibodies (Table 3) overnight at 4 °C. TBS-T buffer instead of antibodies was applied as negative control. Secondary horseradish peroxidase-conjugated antibodies were used to detect immunoreactive complexes. In some experiments the membranes were reused (for β-actin detection) after striping with Restore Stripping Buffer (Thermo Fisher Scientific, Waltham, MA, USA). The protein bonds were visualized by enhanced chemiluminescence with Westar Supernova, ECL substrate for Western blotting (Cyangen, Bologna, Italy). The intensity of the bands was quantified densitometrically with the Gene Tools program (Syngene, Frederick, MD, USA) and normalized for β-actin.

To express the relative level of sugar antigen Tn, T, sialyl Tn, sialyl T, Fucα1-3GlcNAc in cell lysates and culture medium, ELISA tests were applied. To assess the expression of carbohydrate antigens, biotinylated lectins (Vector, Burlingame, CA, USA) with high affinity to examined structures (specified in Table 3) with concentration 5 μg/mL were used. Microtiter plates (NUNC F96 Maxisorp, Roskilde, Denmark) were coated with 50 μL of cell lysates or culture medium (100 μg protein/mL) and incubated overnight at RT. Next there was blocking step with 100 μL of blocking reagent for ELISA (Roche Diagnostics, Mannheim, Germany), 1 h at RT. Washing of the wells was performed using PBS with 0.05% Tween (100 μL, three times). Then there was incubation with proper lectins for 2 h at RT. In the next step, 100 μL of horseradish peroxidase avidin D (Vector, Burlingame, CA, USA) was applied to detect carbohydrate antigens. After 1 h incubation (RT), the colored reaction was developed with 100 μL of ABTS (2,2′-azino-bis(3-ethylbenzthiazoline-6-sulfonic acid) (Sigma, St. Luis, MO, USA). Absorbances were read at 405 nm after 20–40 min of incubation at RT. The samples were analyzed in triplicate. One percent BSA, instead of the samples, and washing buffer, instead of lectins or monoclonal antibody, were applied as negative controls.

### 4.6. Statistical Analysis

The obtained results are presented as mean ± SD (standard deviation) from at least three independent determinations. Analyses were carried out using Statistica package (StatSoft, Tulusa, OK, USA). One-way ANOVA, followed by Duncan’s multiple post hoc test, was used to determine statistical differences. Data with *p* > 0.05 were considered as significant.

## Figures and Tables

**Figure 1 ijms-23-08602-f001:**
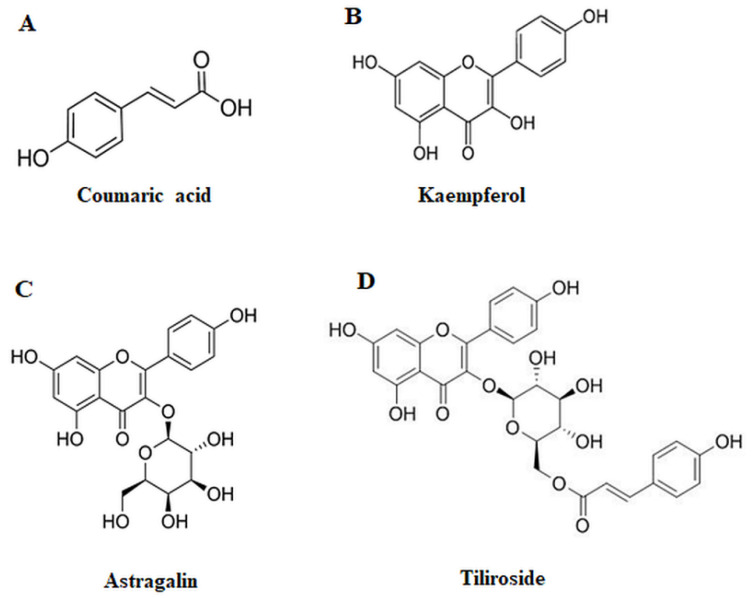
Structures of selected compounds—*p*-coumaric acid (**A**), kaempferol (**B**), astragalin (**C**) and tiliroside (**D**).

**Figure 2 ijms-23-08602-f002:**
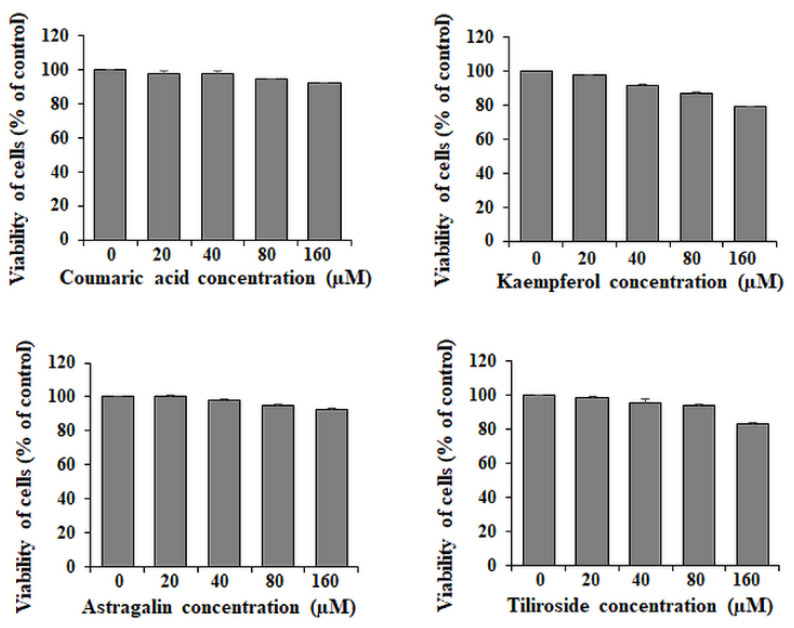
Viability of AGS gastric cancer cells treated for 24 h with 20–160 μM concentrations of coumaric acid, kaempferol, astragalin and tiliroside. Mean values ± SD are the mean of triplicate culture.

**Figure 3 ijms-23-08602-f003:**
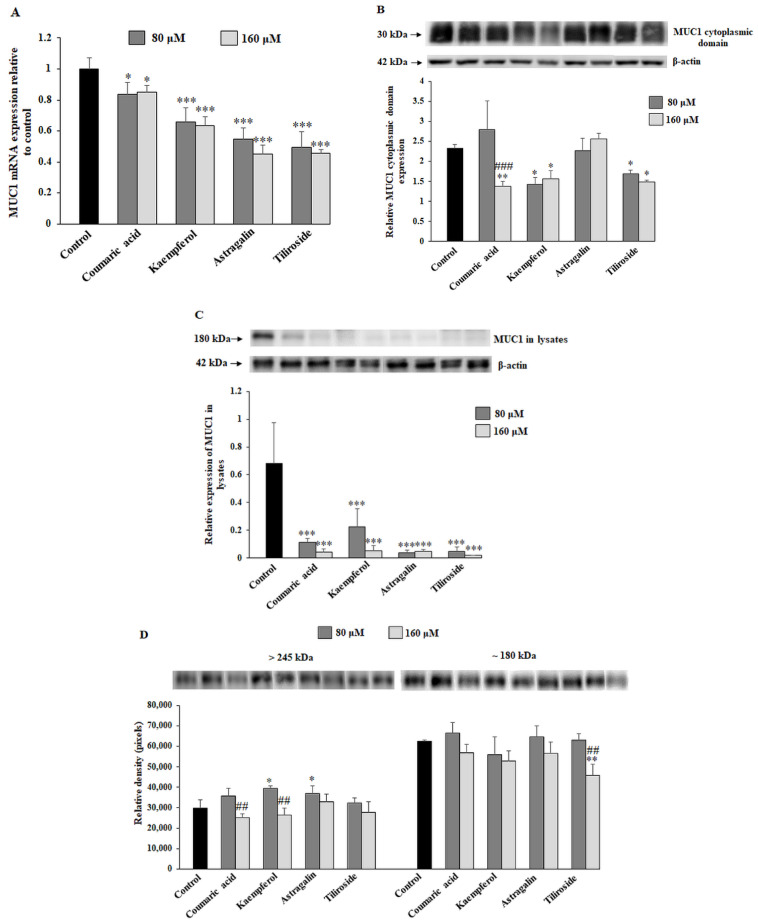
The effect of coumaric acid, kaempferol, astragalin and tiliroside on MUC1 mRNA, MUC1 cytoplasmic domain, MUC1 expression in cell lysates and culture medium. The AGS cancer cells were incubated for 24 h with 80 and 160 μM concentrations of the polyphenolics. mRNA was determined by RT-PCR (**A**). The results are shown as a relative fold change in mRNA expression of gene in comparison to the gene in control, where expression was set at 1. ± SD are the mean of triplicate cultures. * *p* ˂ 0.05, *** *p* ˂ 0.001. MUC1 cytoplasmic domain (**B**), MUC1 in cell lysates (**C**) and in culture medium (**D**) were assessed by Western blot analysis. β-actin served as a protein loading control. The intensities of the bands were quantified by densitometric analysis. Data represent the mean ± SD of triplicate culture. * *p* ˂ 0.05, ** *p* ˂ 0.01, *** *p* ˂ 0.001 compared to untreated control; ## *p* ˂ 0.01, ### *p* < 0.001 compared to lower concentration (80 μM) of specific polyphenolic compound.

**Figure 4 ijms-23-08602-f004:**
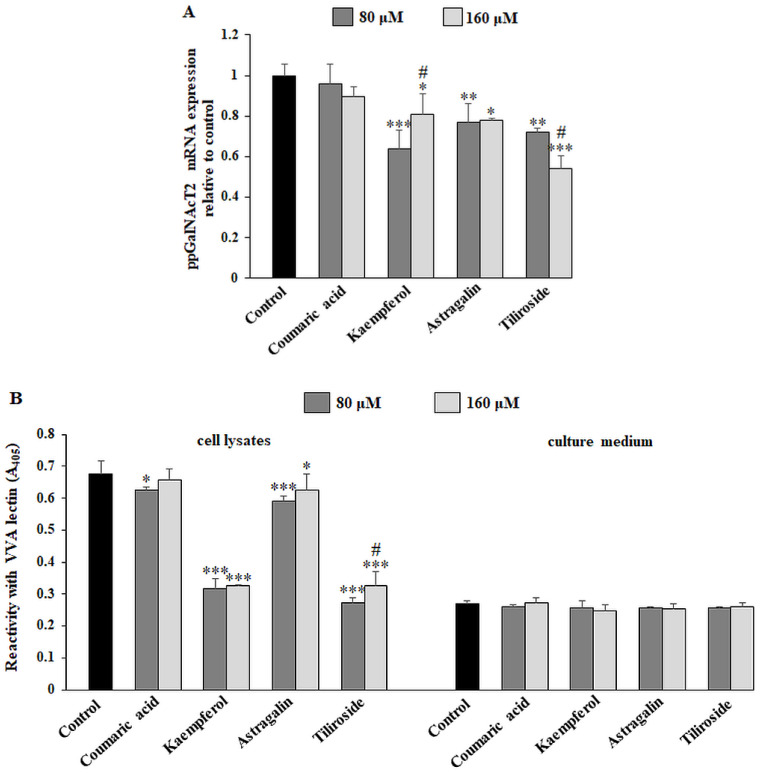
The effect of coumaric acid, kaempferol, astragalin and tiliroside on ppGalNAcT2 mRNA, Tn antigen in cell lysates and culture medium, ST6GalNAcT2 mRNA, and sialyl Tn antigen expression in cell lysates and culture medium. The AGS cancer cells were incubated for 24 h with 80 and 160 μM concentrations of the polyphenolics. mRNA was determined by RT-PCR ((**A**) for ppGalNAcT2 and (**C**) for ST6GalNAcT2). The results are shown as a relative fold change in mRNA expression of genes in comparison to the genes in control where expression was set at 1. ± SD are the mean of triplicate cultures. * *p* ˂ 0.05, ** *p* ˂ 0.01, *** *p* ˂ 0.001 compared to untreated control; # *p* < 0.05 compared to lower concentration (80 μM) of specific polyphenolic compound. Tn and sialyl Tn antigens’ relative expressions in cell lysates and culture medium were analyzed by ELISA tests with biotinylated lectins (VVA recognizing Tn structure (**B**) and SNA recognizing sialyl Tn antigen (**D**)). The results are expressed as absorbance at 405 nm after reactivity with proper lectin. Values ± SD are the mean from three independent assays. * *p* ˂ 0.05, ** *p* ˂ 0.01, *** *p* ˂ 0.001 compared to untreated control; # *p* < 0.05, ### *p* < 0.001 compared to lower concentration (80 μM) of specific polyphenolic compound.

**Figure 5 ijms-23-08602-f005:**
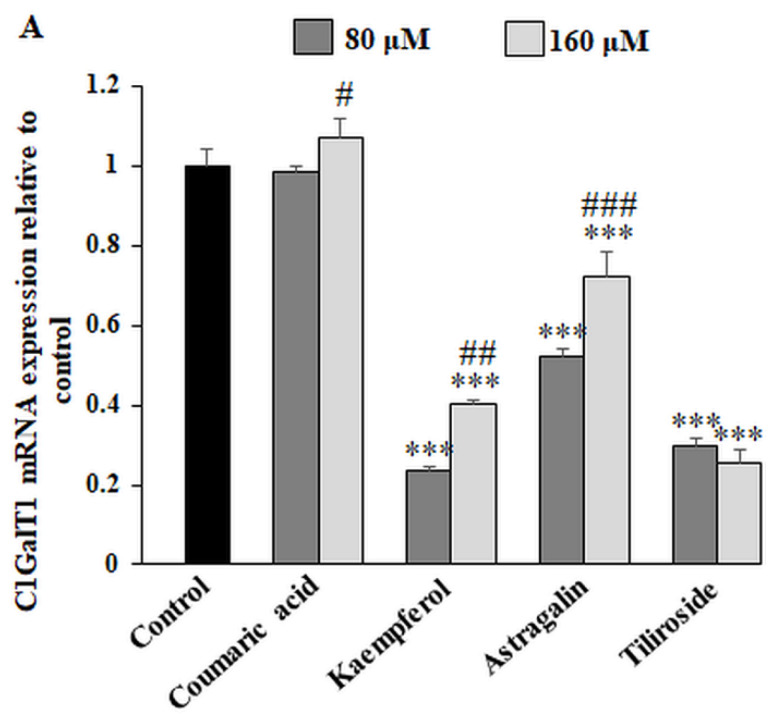
The effect of coumaric acid, kaempferol, astragalin and tiliroside on C1GalT1 mRNA, C1GalT1 protein, T antigen in cell lysates and culture medium, ST3GalT1 mRNA, St3Gal-IV protein and sialyl T antigen in cell lysates and culture medium. The AGS cancer cells were incubated for 24 h with 80 and 160 μM concentrations of the polyphenolics. mRNA was determined by RT-PCR ((**A**) for C1GalT1 and (**D**) for ST3GalT1). The results are shown as a relative fold change in mRNA expression of genes in comparison to the genes in control, where expression was set at 1. ±SD are the mean of triplicate cultures. * *p* ˂ 0.05, ** *p* ˂ 0.01, *** *p* ˂ 0.001 compared to untreated control; # *p* < 0.05, ## *p* ˂ 0.01, ### *p* < 0.001 compared to lower concentration (80 μM) of specific polyphenolic compound. C1GalT1 (**B**) and St3Gal-IV (**E**) protein expressions were assessed by Western blot analysis. β-actin served as a protein loading control. The intensities of the bands were quantified by densitometric analysis. Data represent the mean ± SD of triplicate culture. * *p* ˂ 0.05, ** *p* ˂ 0.01, *** *p* ˂ 0.001 compared to untreated control. T and sialyl T antigens’ relative expressions in cell lysates and culture medium were analyzed by ELISA tests with biotinylated lectins (PNA recognizing T structure (**C**) and MAA recognizing sialyl T antigen (**F**)). The results are expressed as absorbance at 405 nm after reactivity with proper lectin. Values ± SD are the mean from three independent assays. * *p* ˂ 0.05, ** *p* ˂ 0.01, *** *p* ˂ 0.001 compared to untreated control; # *p* < 0.05, ## *p* < 0.01 compared to lower concentration (80 μM) of specific polyphenolic.

**Figure 6 ijms-23-08602-f006:**
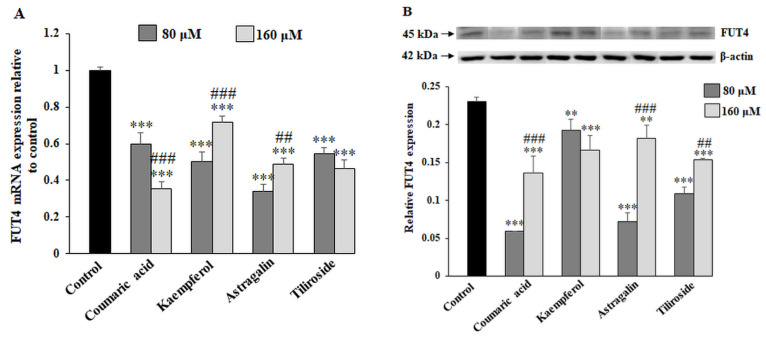
The effect of coumaric acid, kaempferol, astragalin and tiliroside on FUT4 mRNA, FUT4 protein and Fucα1,3-GalNAc antigen. The AGS cancer cells were incubated for 24 h with 80 and 160 μM concentrations of the polyphenolics. mRNA was determined by RT-PCR (**A**). The results are shown as a relative fold change in mRNA expression of genes in comparison to the genes in control, where expression was set at 1. ± SD are the mean of triplicate cultures. *** *p* ˂ 0.001 compared to untreated control; ## *p* ˂ 0.01, ### *p* < 0.001 compared to lower concentration (80 μM) of specific polyphenolic compound. FUT4 protein expression was assessed by Western blot analysis (**B**). β-actin served as a protein loading control. The intensities of the bands were quantified by densitometric analysis. Data represent the mean ± SD of triplicate culture. ** *p* ˂ 0.01, *** *p* ˂ 0.001 compared to untreated control; # *p* < 0.05, ### *p* < 0.001 compared to lower concentration (80 μM) of specific polyphenolic compound. Fucα1,3-GalNAc antigen relative expression in cell lysates and culture medium was analyzed by ELISA tests with biotinylated lectins (LTA recognizing fucosylated antigen) (**C**). The results are expressed as absorbance at 405 nm after reactivity with the lectin. Values ± SD are the mean from three independent assays. ** *p* ˂ 0.01, *** *p* ˂ 0.001 compared to untreated control; # *p* < 0.05, ### *p* < 0.001 compared to lower concentration (80 μM) of specific polyphenolic compound.

**Figure 7 ijms-23-08602-f007:**
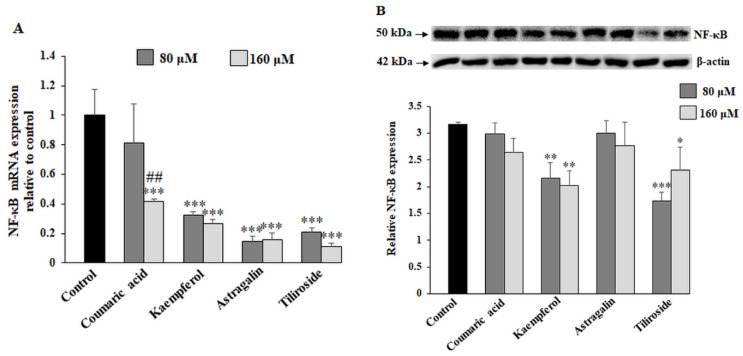
The effect of coumaric acid, kaempferol, astragalin and tiliroside on NF-κB mRNA and protein. The AGS cancer cells were incubated for 24 h with 80 and 160 μM concentrations of the polyphenolic compounds. mRNA was determined by RT-PCR (**A**). The results are shown as a relative fold change in mRNA expression of genes in comparison to the genes in control where expression was set at 1. ± SD are the mean of triplicate cultures. *** *p* ˂ 0.001 compared to untreated control; ## *p* ˂ 0.01 compared to lower concentration (80 μM) of specific polyphenolic. NF-κB protein expression was assessed by Western blot analysis (**B**). β-actin served as a protein loading control. The intensities of the bands were quantified by densitometric analysis. Data represent the mean ± SD of triplicate culture. * *p* ˂ 0.05, ** *p* ˂ 0.01, *** *p* ˂ 0.001 compared to untreated control.

**Figure 8 ijms-23-08602-f008:**
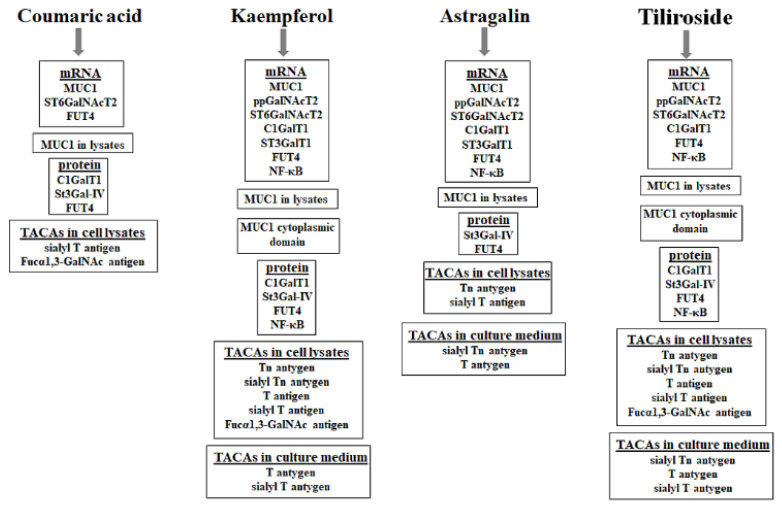
Summarized inhibitory effect of coumaric acid, kaempferol, astragalin and tiliroside on specific glycoforms and NF-κB in AGS gastric cancer cells.

**Table 1 ijms-23-08602-t001:** Lectins used in ELISA tests.

Origin and Abbreviations of Lectins	Binding Preference
*Arachis hypogaea* (peanut) (PNA)	Galβ1,3-GalNAcα1-O-Ser/Thr (T antigen)
*Lotus tetragonolobus* (LTA)	Fucα1,3-GlcNAc
*Maackia amurensis* (MAAII)	NeuAcα2,3-Gal (sialyl T antigen)
*Sambucus nigra* (SNA)	NeuAcα2,6-Gal/GalNAc (sialyl Tn antigen)
*Vicia villosa* (VVA)	GalNAcα1-O-Ser/Thr (Tn antigen)

**Table 2 ijms-23-08602-t002:** Primers used for RT-PCR.

Gene	Forward Primer (5′→3′)	Reverse Primer (5′→3′)
*MUC1*	TGCCTTGGCTGTCTGTCAGT	GTAGGTATCCCGGGCTGGAA
*C1GalT1*	AAGCAGGGCTACATGAGTGG	GCATCTCCCCAGTGCTAAGT
*ppGalNAcT2*	AAGAAAGACCTTCATCACAGCAATGGAGAA	ATCAAAACCGCCCTTCAAGTCAGCA
*ST6GalNAcT2*	CCTTCTGAACGGCTCAGAGAGT	GCACACCGGATACACTTTGGA
*ST3GalT1*	TCGGCCTGGTTCGATGA	CGCGTTCTGGGCAGTCA
*FUT4*	AAGCCGTTGAGGCGGTTT	ACAGTTGTGTATGAGATTTGGAAGCT
*NF-κB*	CTGAACCAGGGCATACCTGT	GAGAAGTCCATGTCCGCAAT
*GAPDH*	GTGAACCATGAGAAGTATGACAA	CATGAGTCCTTCCACGATAC

**Table 3 ijms-23-08602-t003:** Antibodies used in Western blotting.

Antibody	Clone	Source
Anti-MUC1; extracellular domain (mouse IgG) Anti-MUC1; cytoplasmic tail (Armenian hamster IgG) Anti-NF-κB (mouse IgG) Anti-C1GalT1 (mouse IgG) Anti-FUT4 (mouse IgG) Anti-St3Gal-IV (mouse IgG) Anti-β-actin (rabbit IgG) Anti-mouse IgG peroxidase conjugated Anti-rabbit IgG peroxidase conjugated Anti-Armenian hamster IgG peroxidase conjugated	BC2 CT2 5D10D11 F-31 A-10 1F4	Abcam Abcam Cell Sign Tech Santa Cruz Santa Cruz Santa Cruz Sigma Sigma Sigma Abcam

## Data Availability

Data available on request. The data presented in this study are available on request from the corresponding author.

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
