# Peer review of "p-Coumaric acid, Kaempferol, Astragalin and Tiliroside Influence the Expression of Glycoforms in AGS Gastric Cancer Cells"

_ijms, 2022, doi:10.3390/ijms23158602_

Round 1

Reviewer 1 Report

Title: is not "the expression of glycoforms IN (not of) AGS Gastric cancer cells?"

Fig.1 The resolution of the image is low, moreover is better to place the image in the center and not on the left

Fig.2 I don't like the resolution and the graph that you use to show the cell viability. 

Why did you choose to treat the cells for 24h for all your treatments? 

Fig. 5 A,B,C,D the character size of 80 and 160 uM is different in the images, please correct. 

Finally, Can you please improve the quality of all images? 

Author Response

Title – has been changed according to suggestion

Fig. 1 – has been improved according to suggestions

Fig. 2 – has been changed according to suggestions. In the current work it is presented in form of bar graphs. 24 h of treatment of the cells with proper compounds is rather typical time. We have applied such scheme of treatment in all our procedures. We state that this period is optimal, longer incubation could be destructive for the cells.

Fig. 5 – corrected according to the suggestion

Generally – we tried to improve the quality of all images.

Reviewer 2 Report

The manuscript by Radziejewska and co-workers addresses the interesting question on how cellular glycosylation may be modulated by external compounds. Most importantly it investigates how some natural products may be used to modulate the expression o shot chain O-glycans commonly associated with cancer aggressiveness. The manuscript is clearly written.

The work is experimentally sound and seems to have been carried out carefully with protein expression validated both at the mRNA and Protein levels. Expression of O-glycan antigens was evaluated by lectin affinity-blotting. Data should be taken carefully as some lectins may present cross-reactivity with other sugar antigens. A mass spectrometry-based O-glycomics approach would be ideal to validate these results.

The evaluation of MUC1 expression also seams a bit disjointed from the rest of the work. It would be most interesting if the authors could demonstrate that this protein is actually one of the main carriers of the identified O-glycans. This could be achieved easily by characterizing post-translational modifications by mass spectrometry in the electrophoretic band of MUC1. Or by immuneprecipitating the protein and cross-reacting in lectin-blotting.

Never-the-less the work is only descriptive and fails to co-relate o-glycan or MUC1 expression with cellular adhesion, invasion capacity, resistance to therapeutic options or other feature of cancer aggressiveness. Being able to achieve this goal would bring a much higher relevance to what are now interesting descriptive results.

Author Response

Lectin affinity-blotting – we agree that lectins are not fully specific to one sugar structure and cross-reactivity is possibly. However, based especially on the literature, we assumed the highest affinity of lectins to the structures mentioned in the table 3. A mass spectrometry-based O-glycomics approach seems to be excellent method to validate our results. I hope that it will be possible in our future work that is planned as continuation and extension of current research.

Evaluation of MUC1 expression – we also agree that mass spectrometry in the electrophoretic band of MUC1, immunoprecipitating the protein and cross-reacting in lectin-blotting would be valuable method to validate that MUC1 is main carrier of the identified O-glycans. Upon our current results we can’t state this, we can only assume. And again I believe that such idea will be possible to be realized in our future experiments as mentioned above.

Experiments concerning cellular adhesion, invasion capacity, resistance to therapeutic options and other features of cancer aggressiveness, in the presented subject, will be also planned.

Thank you again for such valuable suggestions.

Reviewer 3 Report

Iwona Radziejewska and colleagues in the article p-Coumaric acid, Kaempferol, Astragalin and Tiliroside Influ- 2 Infence the Expression of Glycoforms of AGS Gastric Cancer Cells 3 have shown that they can control the abnormal glycosylation of cancer cells that typically cause the development of carcinogenic conditions specifically in gastric cancer through substances these natural substances. The article has been well organized and the figures are adequate and well described. In the discussion to improve the reading one can add a general scheme of how p-Coumaric acid, Kaempferol, Astragalin and Tiliroside Influ-2 work which is described in the text.

Author Response

We added a general scheme summarizing all inhibitory results (as Fig. 8) to the Discussion section – according to the Reviewer’s suggestion.

Round 2

Reviewer 2 Report

The authors have decided to ignore my suggestions to improve the overall significance/impact of the manuscript. Although stating in the author’s reply letter that they agree with some of the raised comments, they fail to incorporate any of this concerns as additional experiments of in the paper’s discussion. Being so, I feel the manuscript lacks significance for publication in this forum, and I do not feel comfortable to support its publications as it is.

Author Response

Response to Reviewer 2:

We are really very sorry that our answer has been understood in a such way that the Reviewer wrote. Definitely it was not our intension. We want to apologize that we absolutely didn’t want to ignore the reviewer’s valuable suggestions concerning improving of the manuscript. We appreciate all the suggestions very much. Unfortunately they are not possible to be introduced now to the current work. For us such procedures need time. The cell cultures have to be repeated, the workshop should be carefully prepared, the suggested procedures have to be adjusted to the specific conditions concerning methods.

We must emphasize that all of them will be performed in the future work as continuation of the subject. Now we agree that we should add some points concerning Reviewer’s valuable suggestions into Discussion section (this was introduced in the newest version of our work – seen in red in the last part of the Discussion).

We still believe that our outcomes (even in the current for) are worth to be published. We state our work as novel because according to our knowledge there are limited studies concerning the effects of mentioned polyphenolics on glycosylation of gastric cancer cells. We can add that at this stage the work can be treated as kind of preliminary one. As it was mentioned above, the subject will be continued, expanded with suggested procedures in the future.

We conclude that in our opinion the presented outcomes regarding usage of examined compounds seem to be promising as potential anti-cancer treatment.

Round 3

Reviewer 2 Report

I believe the manuscript can be accepted in its current form.

Author Response

Thank you very much for your answer. Again I want to emphasize that we appreciate your comments and treat them as valuable suggestions in the continuation of our work.